# Graduates' Opium? Cultural Values, Religiosity and Gender Segregation by Field of Study

**Izaskun Zuazu**

Institute for Socio-Economics, University of Duisburg-Essen, 47057 Duisburg, Germany;
izaskun.zuazu-bermejo@uni-due.de

**Abstract:** This paper studies the relationship between cultural values and gender distribution across fields of study in higher education. I compute national, field and subfield-level gender segregation indices for a panel dataset of 26 OECD countries for 1998–2012. This panel dataset expands the focus of previous macro-level research by exploiting data on gender segregation in specific subfields of study. Fixed-effects estimates associate higher country-level religiosity with lower gender segregation in higher education. These models crucially control for potential segregation factors, such as labor market and educational institutions, and gender gaps in both self-beliefs and academic performance in math among young people.

**Keywords:** horizontal gender segregation; higher education; cultural values; religiosity; math beliefs; association index

---

## 1. Introduction

Women currently outnumber men in virtually all higher education systems in Western countries. Nevertheless, women and men are strikingly concentrated in specific fields of study. This horizontal gender segregation in higher education results in the over-representation of women in some specific fields (generally in care and humanistic-related fields) and the over-representation of men in others (generally, in technical and science-related fields) (Barone 2011).

Horizontal gender segregation in education is considered an issue of first-order importance insofar as it shapes the skill composition of the future workforce (Altonji et al. 2015) and thus may represent a hurdle for labor market productivity gains and economic development (Dollar and Gatti 1999; Knowles et al. 2002). Furthermore, gender segregation in education accounts for a notable share of the gender wage gap (Brown and Corcoran 1997; Blau and Kahn 2000; Bobbitt-Zeher 2007). Indeed, the female shortfall in the fields of science, technology, engineering and math (STEM) has recently attracted the attention of scholars (Sassler et al. 2017; Card and Payne 2017; Kahn and Donna 2017) and is a major concern in educational and labor market policy-making (Burchell et al. 2014; Figures 2012).

Social scientists from different disciplines point to socio-economic factors, such as gender differentials in career and family aspirations, gender-based discrimination and cultural values as major causes of horizontal gender segregation in education (Ceci et al. 2014). Yet, the theories of horizontal gender segregation in education have not been systematically examined using actual trends (Mann and DiPrete 2013). Bertrand (2017) argues that the scarcity of women on particular educational tracks might be partly driven by constraints expected by women in the jobs associated with those tracks, and highlights the need for further research to help understand the full set of determinants of current gender disparities in educational outcomes.

This paper seeks to close this gap by focusing on the role that cultural values play in horizontal gender segregation in higher education from a cross-country time-series econometric approach. Anti-egalitarian gender attitudes have previously been found to slowdown gender convergence in labor

market outcomes (Fortin 2005). The literature also associates religiosity with more traditional gender roles and less favorable attitudes towards working women (Guiso et al. 2003; Algan and Cahuc 2006). These accounts motivate the current paper to assess the impact of two focal cultural values, namely gender-egalitarian social norms and levels of religiosity, on the gender distribution of higher education graduates across fields of study.

To map segregation trends, I combine national-level measures of gender segregation with disaggregated indices of gender segregation in 9 fields and 23 subfields of study for a panel of 26 Organization for Economic Co-operation and Development (OECD) countries for 1998–2012. This combination of data allows us to uncover patterns of gender segregation that remain concealed when aggregate data on higher education are used. Hence, I am able to identify the precise fields and subfields that drive national-level gender segregation. Cases in point are of agriculture, a generally male-dominated field made up of a highly male-dominated subfield (*agriculture, forestry and fishery*) and a highly female-dominated sub-field (*veterinary*), among other fields of study.

I link the data on horizontal gender segregation in higher education with information on two focal cultural traits: gender equality and religious beliefs. I measure country-level gender equality by means of either the Gender Equality index of the International Institute for Democracy and Electoral Assistance (IDEA). I use the level of religiosity obtained from five waves (1990–1994; 1995–1998; 2000–2004; 2004–2009; 2010–2014) of the World Value Survey (WVS) as a measure of the extent to which social norms are attached to traditional gender roles (Inglehart et al. 2014).

To isolate the impact of cultural values, I control for economic structural changes, labor market and education system features, along with marriage market indicators, such as fertility and divorce rates, as potential determinants of gender disparities in education choices. Finally, I attempt to control for gender gaps in academic performance and self-reported math beliefs among young people that might relate to choices at later stages of their education (Ceci et al. 2014; Eccles and Wang 2016). I use two waves of survey data (2003 and 2012) collected from the Program for International Student Assessment (PISA) to construct aggregate indices of gender differences in anxiety, self-concept and self-efficacy towards mathematics.

The main finding suggests that there is a significant relationship between religiosity and lower levels of gender segregation. The indices of gender equality or inequality are not found to be significantly related to horizontal gender segregation. Gender gaps in math beliefs among young people are found to be correlated with higher gender segregation, which hints at an important link between attitudes acquired in early stages of a lifetime and later education choices. Field and subfield-specific analyses provide a bigger picture of these correlations. The disaggregated results suggest that religiosity might be conducive to lower gender segregation in the fields of agriculture and health and welfare, and more specifically in the subfields of mathematics and statistics, agriculture, forestry and fishery and social services.

The remainder of the paper is structured as follows. Section 2 provides reasons for considering a link between culture and gender segregation. Section 3 describes the data. Section 4 specifies the empirical strategy. Section 5 shows national, field and subfield-level results. Section 6 concludes.

## 2. Gendered Choices of Field of Study

Standard economic literature considers the choice of major as a dynamic process of decision-taking under uncertainty in which individuals make assumptions so as to infer the outcomes of their specific, field-of-study choices (Altonji 1993; Arcidiacono 2004; Zafar 2013). Those assumptions may include neoclassical economic explanations such as foreseen family burdens and discrimination to explain gender disparities in education choices[1]. Experimental economics, for its part, seems to debunk the often-repeated arguments of innate gender differentials in cognitive skills by showing that gender gaps in risk-taking, competitive-leaning and social beliefs drive gendered choices of fields of study (Croson and Gneezy 2009; Buser et al. 2014).

Parallel to these explanations, economic research on cultural values emphasizes the role of gender identity and social norms in shaping the economic behavior of people (Guiso et al. 2006; Blau et al. 2013; Giuliano 2017) (see footnote 1). The shift from traditional to egalitarian social norms regarding gender roles has paved the way towards gender convergence in educational investment and labor market outcomes (Fortin 2005; Mandel and Semyonov 2006). On this bedrock of cultural values, Guiso et al. (2003) affirms that religion is likely to affect every aspect of life in society. Using World Value Survey data, they associate religiosity with less favorable attitudes towards working women. Algan and Cahuc (2006) assess the attachment of religion to traditional family values that favor a male breadwinner division of labor. They document differences between religion denominations, in which Catholics and Muslims are more likely to agree with traditional gender role prescriptions than Protestants or non-religious people. Based on these different prescriptions on the role of working women across societies, one might consider that culture can either encourage or hinder gender divergence in the choices of major in higher education.

The epidemiological methodology developed in Fernández (2011) reinforces the explanatory power of the intergenerational transmission of gender norms on gender disparities in both individual and constrained preferences in the labor market and educational choices (Farré and Vella 2013; van de Werfhorst 2017; Charles et al. 2018). However, the role of culture has not been addressed in international comparisons of horizontal gender segregation in education in depth due to scarcity of data available. Drawing on the empirical evidence supporting the idea that economic outcomes and social beliefs are correlated,[1] the canonical arguments of gender segregation are framed in rational choice theory and are divided into demand-side factors (Mincer and Polachek 1974) and supply-side factors (Becker 1957). For recent research, see Goldin (2006, 2014a, 2014b).

The current paper considers whether cultural values (e.g., gender equality and religion) play a role in horizontal gender segregation in higher education. Gender segregation explanations drawn from prior mathematical achievement have been steadily replaced by findings suggesting that gender disparities in perceived ability have stronger effects (Friedman-Sokuler and Justman 2016; Justman and Méndez 2018). Eccles and Wang (2016) use survey data on 1200 college-bound students in Michigan (U.S.) to study whether their self-concept of math ability in 12th-grade (ages 17–18) encouraged them to choose STEM occupations at age 29. Their results indicate that gender differences in the likelihood of entering STEM careers were strongly predicted by math self-concept, together with lifestyle expectations, demographics and high school course-taking, rather than by actual math performance. In a similar vein, Shi (2018) uses data in the transition from high school to college for North Carolina (U.S.) to study female under-representation in engineering. She finds that the scarcity of women in engineering is partly explained by their relative lack of confidence in math abilities, but she finds gender disparities in preferences and professional goals to have stronger explanatory power. Ultimately, these analyses disentangle the segregative effects of sex differences in preferences and aspirations from those arising from disparities in math performance and math-ability perceptions.

This paper adopts a macro-level approach grounded on two earlier works on gender segregation across fields of study: First, the paper by Charles and Bradley (2009), which uses a cross-country analysis of gender segregation in fields of study for 44 countries in 1999; and second, the panel data analysis of US graduates in 225 fields between 1975–2002 by England and Li (2006). I depart from these previous papers by conducting a panel data analysis of gender segregation at national, field and subfield levels and focusing on cultural values while using more nuanced measures of gender gaps in math beliefs. Hence, my approach is intended to tackle both within-country time dynamics of segregation and by-subfield heterogeneity within gender-dominated fields (e.g., *veterinary*

---

[1]   Akerlof and Kranton (2000) provide a game theoretical model that defines an identity-based utility of individual choices. Obeying social prescriptions of one's identity as a "man" or as a "woman" is rewarded while violating them evokes anxiety and discomfort. Hence, this model defines non-pecuniary benefits derived from the choice of educational paths, as formulated for instance by Humlum et al. (2012) and Beffy et al. (2012).

versus *forestry* within agriculture). Due to data limitations, I can only test macro-level relationships between cultural and horizontal values. Cohort-data research finds gender differentials in education outcomes on the basis of demographics, such as immigration (Alonso-Villar et al. 2012), socioeconomic status (Bailey and Dynarski 2011; van de Werfhorst 2017), parents' educational attainment and labor market participation rates (Fernández 2011; Farré and Vella 2013), role models of teachers and parental expectations (Bettinger and Long 2005; Xie and Shauman 2003) and peer-related processes (Schoon and Eccles 2014). The potential intersection between gender and demographics is left for future research.

## 3. Data on Gender Segregation

The OECD Education Database classifies the number of female and male graduates based on the International Standard Classification of Education (ISCED1997) in 9 broad fields of study (1 digit-level) and 23 narrow fields of study (2 digit-level), which I refer to as subfields (see Table A1 in Appendix A). I collect data for 26 OECD countries for 1998–2012[2]. The selection of these countries corresponds to the availability of data and the quality thereof, which is the highest coverage possible using OECD countries. Using data on graduate completion instead of enrolment rates mitigates issues of attrition in gender-atypical choices, specifically in female students (Mastekaasa and Smeby 2008). To the best of my knowledge, this data allows for the greatest country coverage, time span and data disaggregation to compute gender segregation indices in the context of Western countries, although some limitations in the data collection and harmonization across countries might be expected. I use two nominal measures of gender segregation: The Dissimilarity Index (Duncan and Duncan 1955) and the Association Index (Charles and Grusky 1995). The former provides information at the national-level and the latter at field or subfield-levels of segregation[3].

### 3.1. Country-Level Segregation: Dissimilarity Index

The index of dissimilarity (ID hereafter) was first developed in racial segregation studies by Duncan and Duncan (1955). The ID is one of the primary measures of segregation applied to the context of gender segregation in labor markets and education (Gelbgiser and Albert 2018). It is given by the following formula[4]:

$$ID = \frac{1}{2} \sum \left| \frac{Fi}{F} - \frac{Mi}{M} \right| * 100 \tag{1}$$

where $F_i$ and $M_i$ are females and males in field or subfield $i$, $F$ and $M$ are the total numbers of female and total male graduates, respectively. As defined in Duncan and Duncan (1955), the ID provides the percentage of women who would have to change fields without replacement in order to make their distribution identical to that of men. The index takes values from 0%, indicating total gender integration across fields, to 100%, indicating complete gender segregation.

Figure 1a shows the trend of the sample average ID computed based on broad (ID at 1 digit-level, blue line) and narrow (ID at 2 digit-level, red line) classifications of higher education. The ID is sensitive to the techniques and categorizations used in defining fields (Reskin 1993; Nelson 2017). Consequently, the ID can be manipulated into being smaller (by using very broad categories) or larger (by using

---

[2] See Andersson and Olsson (1999).

[3] In sharp contrast to ordinal measures, nominal measures of segregation do not take into account a hierarchical ordering of the education system (Semuonov and Jones 1999). A large body of American literature on the pay-offs to human capital suggests that generally female-dominated fields (humanities and social science) result in lower incomes than male-dominated fields (scientific and technical fields) (Charles and Bradley 2009). Nevertheless, given the lack of specific data on wages associated with each field or subfield for the sample of countries, the current paper does not distinguish between female and male-dominated fields in any income or social status ordering.

[4] Cross-national and inter-temporal comparisons using the ID might entail computational issues due to its sensitivity to the share of fields in total higher education (Charles and Grusky 1995; Watts 1998). If education systems are dominated by one highly segregated field, the ID would yield higher values than if the dominant field was evenly composed by women and men, and numerous small fields were highly segregated.

narrow categories). This sensitivity is evident in the different average levels taken by the ID in broad or narrow categorizations (disaggregation at 1 digit-level vs. 2 digit-level), where the latter give higher figures for segregation. Regardless of the category used to compute the ID, the indices show a decreasing trend in 1998–2012, with a drop of around 3 percentage points (pp) by the end of the period. However, ID values remain quite stable throughout this period in comparison with de-segregative fashion taken from 1970 to 1990 (England and Li 2006; Mann and DiPrete 2013; Bronson 2015). This might feed into the afore-mentioned slowdown in gender integration in higher education and other areas of society since the mid-1990s (see inter alia Blau et al. (2006); Olivetti and Petrongolo (2016)).

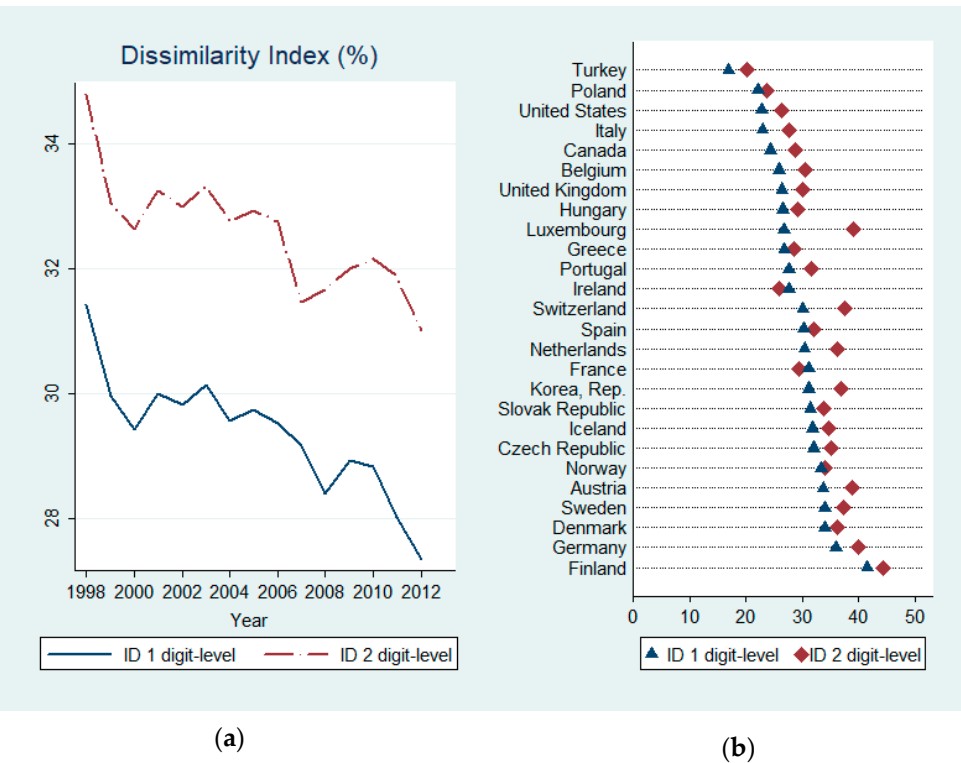

(**a**)　　　　　　　　　　　　　　　　　　(**b**)

**Figure 1.** Country-level Horizontal Gender Segregation. (**a**) Trend; (**b**) By country.

Figure 1b shows average levels of the ID computed at 1 digit-level (blue) and 2 digit-level (red) for each country in the sample. Turkey is the least segregated country in the sample (an ID of 17.1% at 1 digit-level), whereas Finland is the most segregated (42.1%). Cross-country comparisons show that more affluent, more gender-egalitarian countries have greater segregation (e.g., Scandinavian countries[5]).

Current cross-national comparisons challenge rational choice theories that predict less segregation as societies become economically richer and gender egalitarian (see Estevez-Abe (2005)). Economists note that gender disparities that do not clearly define hierarchical structures relative to vertical disparities are less easily undermined (Goldin 2006; Shavit 2007). Thus, horizontal segregation in higher education can reconcile gender-egalitarian and gender-essentialist values to a greater extent[6]. Indeed, this conundrum is already formulated as the *education-gender-equality* paradox in other social science disciplines (Stoet and Geary 2018), as gender-essentialism has been already explored in the context of choice of majors (Ochsenfeld 2016).

---

[6]　This logic corresponds to "separate-but-equal" gender beliefs as a cause of persisting horizontal gender segregation as suggested by Charles and Bradley (2009) and England (2010).

### 3.2. Field and Subfield-Level Segregation: Association Index

I combine the data on country-level gender segregation with data on field-level segregation and subfield-level segregation. To that end, I use the log-linear modeling approach from Charles and Grusky (1995), namely the Association Index ($A_i$ henceforth), which provides the factor at which each field or subfield of study is associated with a gender (female or male)[7]. The $A_i$ index is computed as follows[8]:

$$A = ln\frac{Fi}{Mi} - \left\{\frac{1}{j} * \sum ln\left(\frac{Fi}{Mi}\right)\right\} \tag{2}$$

where *ln* is the natural logarithm, *j* is the number of fields (this number is 9 when the ISCED1997 1 digit-level is used and 23 for the ISCED1997 2 digit-level), $F_i$ is the number of women in field *i* and $M_i$ is the number of men in field or subfield *i*. Positive values of the $A_i$ indicate that the field is associated with women, near zero values indicate gender-neutrality, and negative values that the field is associated with men. A well-suited feature of the association index is that it compares the extent of segregation of male-dominated and female-dominated fields or subfields.

Figure 2 shows the average factor of gender-labeling of fields ordered from most male-dominated to the most female-dominated for 1998–2006 and 2007–2012. *Engineering* is the most segregated field that happens to be male-dominated, showing an $A_i$ of −1.5. *Science and agriculture* are also male-dominated, although to a lesser extent than *engineering*. Fields placed in the middle of the table, with values around zero, are gender-neutral fields (services and social sciences). *Humanities and arts* comprises a female-dominated field, with values close to 0.5. Finally, *education* and *health and welfare* are the most female-dominated fields with values around 1, and are the most segregated fields after engineering.

Figure 2 also shows that the gender labeling of fields remains similar before and after the Great Recession, although agriculture and humanities are slightly less segregated and science is more segregated on average in 2007–2012. This descriptive data is consistent with the care-technical and humanistic-scientific divides highlighted in Barone (2011).

Figure 3 reveals high heterogeneity in gender-labeling within fields of study. The field of engineering is divided into three subfields with varying factors of gender-labeling: *Manufacturing* is slightly male-dominated, with an index close to zero (−0.16), whereas *engineering* and *architecture* are more male- dominated with values of −1.7 and −0.83. The overall male-dominated fields of science and agriculture have also female-dominated subfields, such as *life science* and *veterinary* studies. Similarly, the field of services is made up of highly male-labeled subfields (*transport services* and *security services*) plus a female-labeled subfield (*personal services*). The most segregated subfields are *engineering* (male-labeled) and *social services* (female-labeled). Averages from before-and-after the Great Recession show that *computing* and *veterinary* are more segregated in 2007–2012, whereas *security services* and *personal services* are less segregated in this latter period.

---

[7] See Charles and Bradley (2009); Barone (2011) and Mann and DiPrete (2013) for applications of the index in the context of segregation in education. Following the sociological literature in which this index was developed, I use the term of "gender-labeling" of fields, although the term "gender-typing" is also used in the literature.

[8] The *Ai* index outperforms ID in cross-country and inter-temporal comparisons as by using log-linear techniques the index is unaffected by the weight of each field in different countries or weight of women. See Watts (1998); Blackburn et al. (1993) for these computational issues of segregation indices.

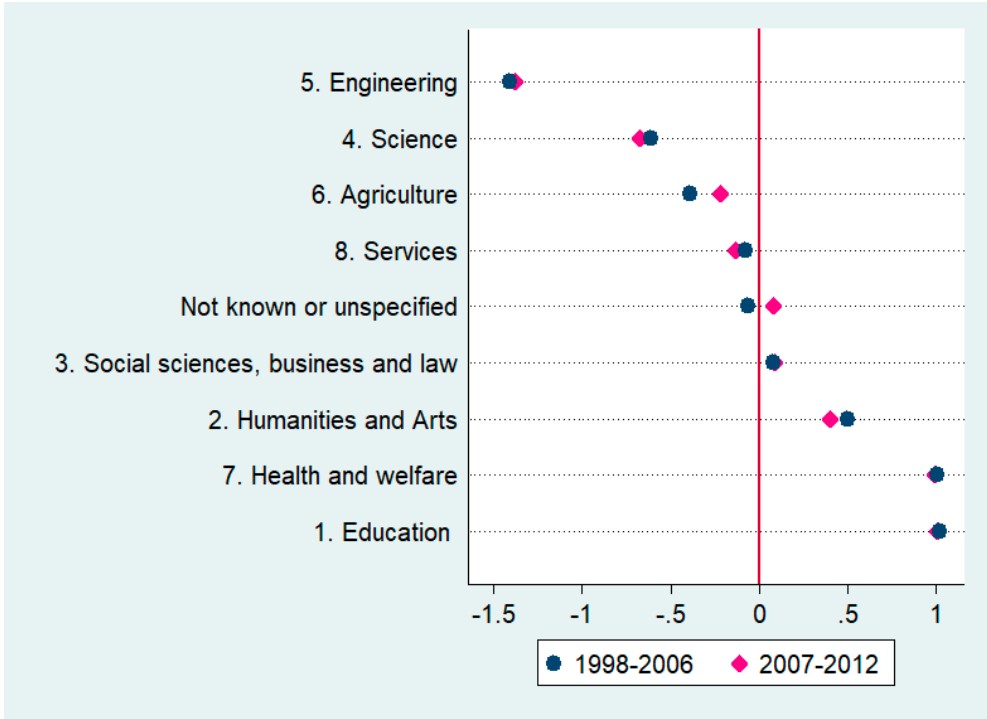

**Figure 2.** Field Segregation—Association Index.

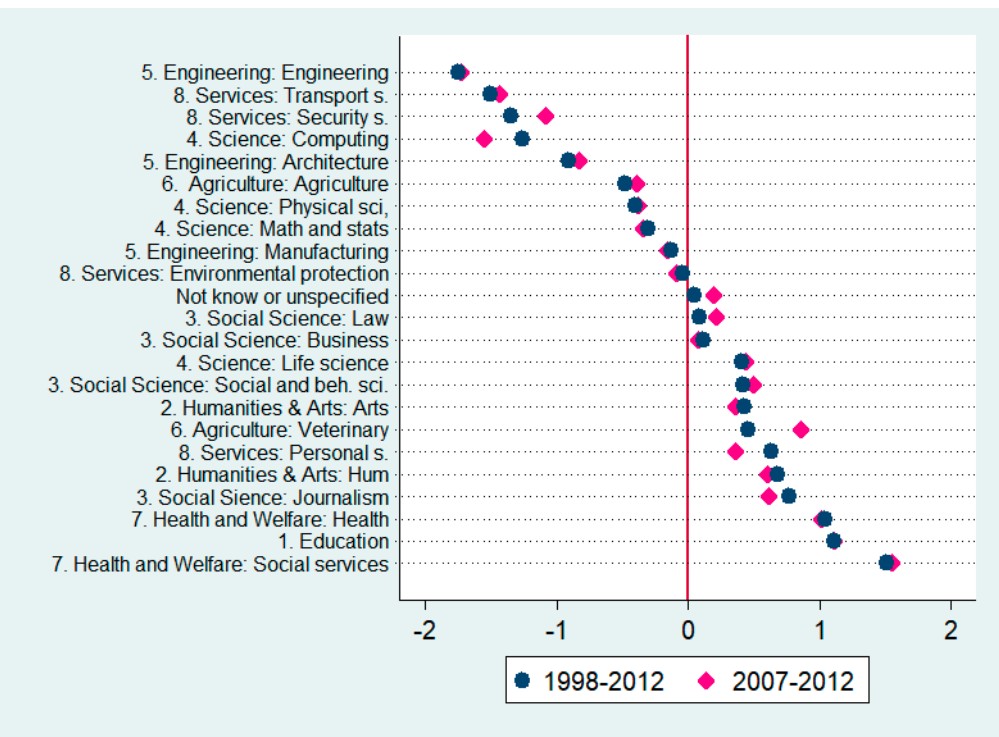

**Figure 3.** Subfield Segregation—Association Index.

## 4. Empirical Strategy

The hypothesis that I test is whether cultural values play a role in the gender distribution across fields of study in higher education. I first specify panel data regression models using country-level

segregation (ID) as the left-hand-side (LHS) variable, computed using either a broad (field) or narrow (subfield) classification of higher education.

$$ID_{ct} = \beta_0 + \beta_1 CulturalValues_{c,t-4} + X^J_{c,t-4}\beta_2 + \gamma_t + \alpha_c + u_{ct}$$
$$c = country; t = year$$
(3)

where $ID_{ct}$ is the dissimilarity index in country $c$ in year $t$, $\gamma_t$ and $\alpha_c$ are time and country fixed-effects, respectively. $CulturalValues_{c,t-4}$ is the focal explanatory variable referring to either country-level gender equality or religiosity. $X_{c,t-4}$ is a set of control variables. Following England and Li (2006), I lag the full set of independent variables four years behind the dependent variable to alleviate causality issues. Considering that the data covers all types of higher education graduates (2-year college, bachelor's degree, master's degree and Ph.D.) a time span of four years to completion might be reasonable. Using lags of proxies for cultural values, like religiosity, can alleviate reverse causation between education and religiosity[9]. I am aware of the difficulty of interpreting the results below as causal effects, so I follow the literature to ease the exposition of the results by talking about "impacts" or "effects". Provided the difficulty of solving for multiple sources of endogeneity common to fixed-effects panel data models (*inter alia*, variable biased, reverse causation), the reader should interpret the results below as mere correlations. Baseline models are computed based on information for 26 countries, although the sample of countries is reduced to 18 when WVS data are used and to 17 for PISA data (see summary statistics and sample countries in Table A2, Appendix A).

*4.1. Measures of Cultural Values: Gender Equality and Religiosity*

I measure gender-egalitarian values by employing the IDEA Gender Equality index. This index is operationalized using five indicators: Power distribution by gender, female participation in civil society organizations, the ratio between mean years of schooling for women and men, the proportion of lower chamber female legislators, and the proportion of women in ministerial- level positions (Skaaning 2017)[10].

I measure country-level religiosity using five waves (1990–1994; 1995–1998; 1999–2004; 2005–2008; 2010–2012) of the WVS. As in the reference literature (Guiso et al. 2003), religiosity is measured by the proportion of WVS respondents who, on a 0–10 scale, give a score of 10 for the statement *"God is very important in my life"*. This statement is present in all WVS waves, whereas other religion-related WVS questions were asked in fewer waves. Average values for Gender Equality and religiosity by country can be found in Figures A1 and A2 shows the evolution of the sample mean of these covariates over time (Appendix C).

Gender-unequal cultural values are thought to reinforce gender-essentialist ideals, i.e., widely shared beliefs that women are better at caring, nurturing and human interaction, whereas men excel at abstract thinking, problem solving and analysis (Sikora and Pokropek 2012; Charles et al. 2015). Anti-egalitarian values might be expected to shape gendered identities of individual men and women to encourage the choice of gender-confirming fields of study, and thus increase segregation. At the same time, the evidence states that more religious ideologies go in lockstep with traditional division of labor and gender roles, which might lead to the expectation of higher levels of segregation in more religious societies.

Figure 4 challenges this view by showing scatter plots of the three alternative proxies of cultural values and gender segregation at national level (dissimilarity index) in 2012. Religiosity is negatively correlated with segregation, meaning that in less gender-egalitarian and more religious societies,

---

9    Dilmaghani (2019) founds a causal link between education and religious unaffiliation. Applying this evidence to the case of gender segregation, one might consider the extent to which the advancement of women in higher education might lead to lower religiosity levels in coming generations.

gender segregation is lower. By contrast, greater gender equality is positively correlated with horizontal gender segregation in higher education.

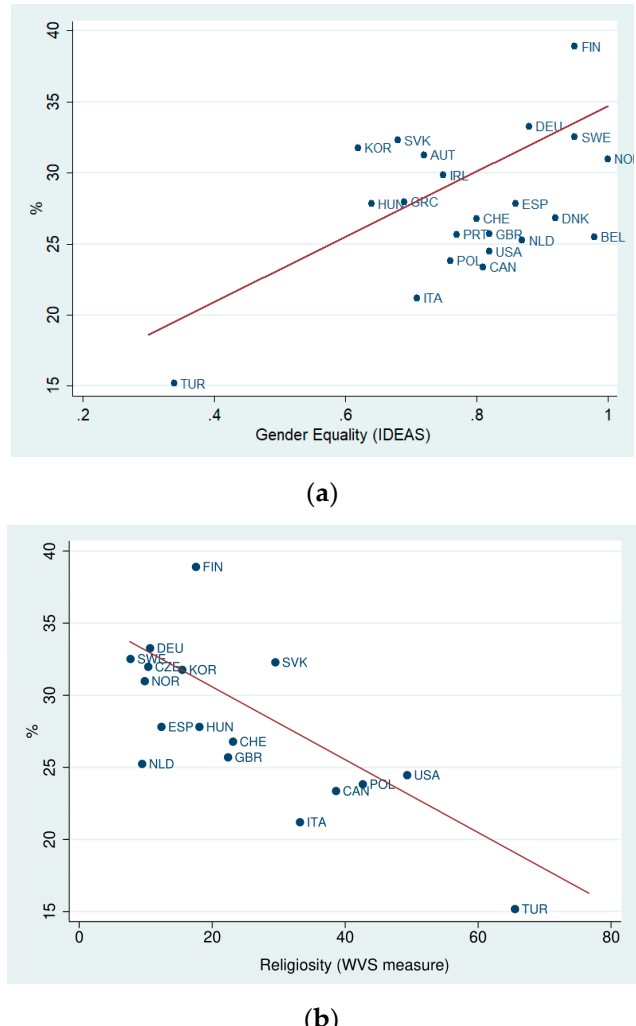

**Figure 4.** Gender Segregation and Cultural Values. (**a**) Gender Equality, (**b**) Religiosity.

*4.2. Control Variables*

The term $X_{c,t-4}$ is a vector of variables measuring economic, labor market and educational institutions that previous literature has related to horizontal gender segregation. These variables are the population density (Pop. density), the share of employees in the service sector to total employment (% Services) and the female labor force participation rate (Female Labor Force). At the same time, I include the percentage of professionals who are female (% Prof. Fem). Regarding educational features, the models include the number of graduates as a proportion of the total population (Size Grads), the percentage of women in the total graduate student body (% Grad. Fem) and the breadth of vocational education via the number of graduates in ISCDE1997 level 5 Type B as a proportion of total higher education (Diversification), as well as the gender gap in academic performance (boys' scores minus those of girls) in secondary education (Performance Gap).

The marriage market-related covariates, specifically fertility and divorce rates and gender gaps in math beliefs are also considered. The set of controls also includes fertility and divorce rates. Goldin (2006) argues that these indicators were among the underpinnings of the transformation of women's role in the labor market from a job-focus to a career-design in the aftermath of World War II. They might in turn foster a convergence between men's and women's choices of education paths.

Along these lines, past papers find that gender discrepancies in marriage aspirations and family formation plans to impact on the share of women in math-related and female higher educational attainment (Badgett Lee and Folbre 2003; Ceci et al. 2014; Bronson 2015; Attanasio and Kaufmann 2017). The current paper controls for these marriage market features, supplementing existing international analyses of segregation such as that of Charles and Bradley (2009). Further information on data sources and pairwise correlations of the explanatory variables are relegated to Tables A3 and A4 in Appendix A.

*4.3. Education System and Performance*

Attitudes of Young People towards Math

An important contribution of this paper is the consideration of gender disparities in self-believes related to math skills during early stages of the educational career. I use the 2003 and 2012 waves of PISA surveys, which focused on mathematics (OECD 2013). This in-depth focus provides data on self-reported beliefs regarding math anxiety (measured by means of students' responses about feelings of stress and helplessness when dealing with mathematics), math self-concept (based on students' responses about their perceived competence in mathematics) and math self-efficacy (based on students' perceived ability to solve a range of pure and applied mathematical problems).

PISA assesses these self-reported math beliefs on the basis of strong agreement or agreement on a number of items in each dimension, which are relegated here to Appendix B. I compute gender gaps in national-level indices of math anxiety, self-concept and self-efficacy based on average agreement with the items for each dimension[11]. In virtually all the countries in the sample, girls are more likely to report math anxiety and less likely to report a self-concept of math than boys. As for math self-efficacy, boys generally report higher levels than girls, although there is some heterogeneity depending on the item in question. Based on the sample of countries, girls show higher levels of self-efficacy in items related to equations (first and second order linear equations). However, boys score higher than girls in the rest of the self-efficacy items. I compute gender gaps for these indices based on the gender that shows higher levels of these self-reported math beliefs: Math anxiety gender gaps are computed as girls' indices of math anxiety minus that of boys, whereas gender gaps in self-concept and self-efficacy are computed as boys' indices minus those of girls.

Figure 5 shows average gender gaps between 2003 and 2012 in math anxiety (blue points), math self-concept (red diamonds) and math self-efficacy (green triangles) in the sample of countries, which are listed from lower to higher gender gaps in math anxiety. Switzerland, Norway, France and Canada show the biggest levels of gender gaps in math anxiety, while Poland, Turkey, Portugal and the Republic of Korea show the lowest. Gender gaps in math self-concept are larger than for math anxiety, gender gaps in math self-efficacy are closer to those of math anxiety.

The data displays a pattern in which affluent and more gender-egalitarian countries generally have wider national gender gaps in self-reported math beliefs than less affluent countries, as found by Stoet et al. (2016).

Figure A3 in Appendix D provides scatter plots of ID and gender gaps in self-reported math beliefs. In all three cases, the plots tend to positively correlate wider gender gaps for young people with gender segregation in higher education.

To control for gender gaps in math beliefs, I apply a linear adjustment for 2003–2012 under the assumption of an equal year by year change in math beliefs over that time. Notice that segregation data spans 1998–2012 and using lags of math beliefs would substantially reduce the number of years (2008–2012) and thus of observations. Hence, following previous literature, I study the contemporaneous

---

[11] PISA provides scale indices of self-reported math beliefs measuring the distance from national levels to average of the total sample of countries participating in PISA surveys. It would be misleading to link these scale indices with my database of gender segregation because my panel is unbalanced and only covers a cluster of OECD countries. Thus, I construct aggregate-level gender gaps in self-reported math beliefs instead of using scale indices in OECD (2013).

effect of math attitudes of young people and gender segregation in higher education graduates. This data does not measure the effects of gendered attitudes towards math at individual level, but it enables me to assess to a certain extent whether patterns of gender segregation correspond to aggregate-level gender differences in math anxiety, self-concept or self-efficacy. The approach here seeks to supplement the cross-country analysis in Charles and Bradley (2009), in which they include TIMSS data on disparities in affinity for math between boys and girls.

$$ID_{ct} = \beta_0 + \beta_1 CulturalValues_{c,t-4} + \beta_2 MathBeliefs_{ct} + X^J_{c,t-4}\beta_3 + \gamma_t + \alpha_c + u_{ct}$$
$$c = country; t = year \tag{4}$$

As in Equation (3), $ID_{ct}$ is the dissimilarity index in country $c$ in year $t$, $\gamma_t$ and $\alpha_c$ are time and country fixed-effects, respectively. Note that the model includes contemporaneous *MathBeliefs* (e.g., gender gaps in math anxiety, self-concept and self-efficacy), whereas the rest of independent variables are four years lagged.

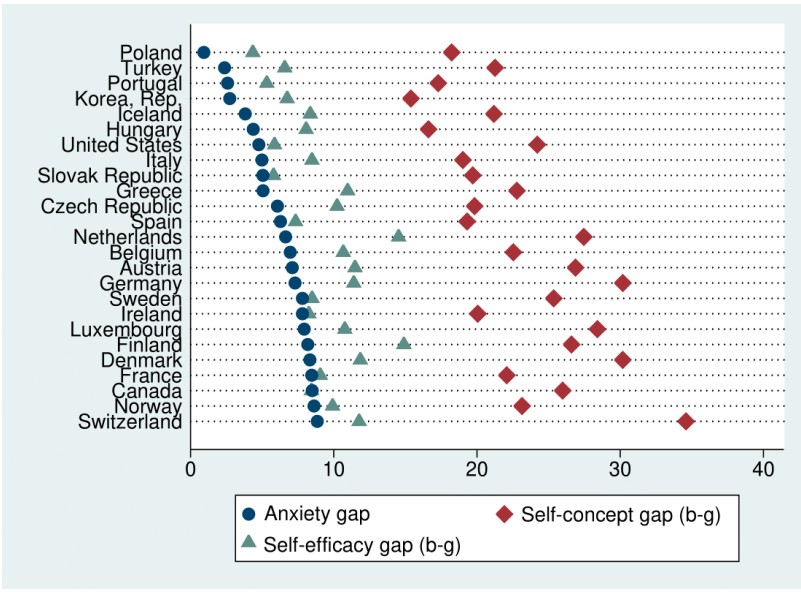

**Figure 5.** Gender Gaps in Self-reported Math Beliefs.

## 5. Results

### 5.1. Country-Level Analysis

I estimate the model in (3) using the within-group estimator. The Breusch and Pagan post-estimation test confirms the presence of conditional heteroskedasticity in the data, so I use cluster standard errors at the country-level and allow residuals to be correlated within but uncorrelated between countries (Cameron and Miller 2015). The Hausman test' initial hypothesis that individual-level effects are adequately modeled by a random-effects model- is resoundingly rejected. Among other post-estimation tests, I take the issue of outliers by identifying observations with very large leverage or squared residuals. I use the lvr2plot Stata command (Cox 2004) to analyze high leverage observations such as those for Turkey and Sweden separately. Excluding these two countries from the sample, the results are unchanged.

A potential caveat on the validity of the estimation is concerned with endogeneity issues arising from the relationship between the ID (dependent variable) and the regressor *%ttrad.Fem*. In separate models, I use the Two Step Least Squared (2SLS) and the number of women in parliaments to instrument the share of female graduates (see Stockemer and Byrne (2011) for a justification of this instrument), and corroborate the main results of the paper. Indeed, post-estimation tests of the

2SLS approach fail to reject the hypothesis that the proportion of females in the graduate body is an exogenous covariate.

Table 1 shows that greater religiosity is associated with lower gender segregation four periods later. Yet the estimates of this effect become less significant when gender gaps in self-reported math beliefs are accounted for. Column 1 estimates a baseline model that includes the main set of control variables. The female labor force variable is associated with a negative impact on segregation, which is consistent with previous research (Ramirez and Wotipka 2001). Increasing female participation in higher education seems to be related to greater segregation by field of study, which is consistent with Charles and Bradley (2009). Nevertheless, that association is not robust to the inclusion of religiosity. The revolutionary indicators in Goldin's parlance, fertility and divorce, are associated with a significant negative and positive effect, respectively, on segregation. The effect of fertility is highly robust and challenges the idea that reducing fertility might foster a convergence between the educational choices of men and women.

Column 33 (Table 1) introduces the Gender Equality index, and it is not associated with the significant coefficient. Column 3 uses instead the level of religiosity, which enters with a negative and significant coefficient. This finding is in line with recent evidence on the link between more traditional societies and greater participation of women in math-related fields (Friedman-Sokuler and Justman 2016), and the findings related to closer gender gaps in math performance in Muslim countries (Fryer and Levitt 2010). Ultimately, this negative correlation suggests that gender is less salient in higher education systems in more religious societies. This finding is consistent with that brought by Falk and Hermle (2018), who use survey data to provide evidence on that higher gender equality favor the manifestation of gender differences in preferences across countries.

I provide two mechanisms to tentatively explain the negative association between religiosity and gender segregation[12]: (i) In more religious societies, women play a traditional role in the labor market (e.g., low female labor force participation rates and high fertility rates). As argued in Bertrand (2017), the constraints and challenges that women expect in the jobs associated with certain education tracks make women reluctant to choose them. Thus, if women expect to play a minor role in the labor market, their choices of majors may be less influenced by these future constraints and they will be more likely to opt for male-dominated education paths (e.g., STEM). (ii) In more religious societies, female participation in higher education is relatively lower. Therefore, those women who do access higher education possess an elite identity that encourages them to transgress gender-confirming norms and opt for male-dominated fields (Charles and Bradley 2009).

I test these potential mechanisms by interacting religiosity with either fertility, the female labor force participation rate or the proportion of women in the total number of graduates in separate models. These interactions are not associated with a significant effect, but estimates on the constitutive terms remain similar to the additive model in Equation (3). Due to the limitations of macro-level data used here, it goes beyond the scope of this paper to go further in these explanations.

Column 4 studies whether different religion denominations explain gender segregation by using four waves of WVS data on the proportion of Catholics, Muslims, Protestants and Jews as the main religion denominations in the sample of countries, using the percentage of the total WVS respondents over the five waves used here who claim to belong to a specific religion. However, none of them are associated with a significant coefficient. Columns 6–8 show within-group estimates of Equation (4). The results positively associate gender gaps in math beliefs of the youth with gender segregation. Recall that the anxiety index is composed by the girls' index minus that of boys, whereas self-concept and self-efficacy are based on the boys' index minus that of girls. As girls report higher levels of anxiety, the gender segregation of higher education graduates across fields is also higher. Similarly, as boys

---

[12] See Figure A4 in the Appendix A for scatter plots of female participation in the labor force, share of graduates and fertility with religiosity.

surpass girls in their sense of self-concept and efficacy towards math, higher education graduates tend to be more segregated. Note that religiosity is not significant when accounting for math anxiety and self-concept gender disparities (Columns 5 and 6) but it remains statistically significant at the 0.10 level when including self-efficacy (Column 7). Table 2 checks the robustness of these results by estimating Equation (3) using the ID at 2 digit-level as the LHS variable. The results are similar to those found using the ID at the broader level

**Table 1.** Country-level Gender Segregation. Dependent variable: Dissimilarity index (1 digit-level).

| | Baseline | Cultural Values | | | Math Beliefs | | |
|---|---|---|---|---|---|---|---|
| | (1) | (2) | (3) | (4) | (5) | (6) | (7) |
| L4.Pop. density | −0.002 | 0.018 | −0.012 | −0.220 | −0.114 | −0.152 | −0.092 |
| | (0.126) | (0.132) | (0.102) | (0.232) | (0.089) | (0.107) | (0.097) |
| L4.% Services | −0.074 | −0.035 | −0.031 | −0.239 | −0.207 | 0.012 | −0.111 |
| | (0.198) | (0.188) | (0.142) | (0.173) | (0.170) | (0.162) | (0.152) |
| L4.% Prof. Fem. | −0.038 | −0.037 | −0.061 | 0.171 * | 0.003 | 0.082 | −0.077 |
| | (0.072) | (0.080) | (0.098) | (0.095) | (0.077) | (0.105) | (0.100) |
| L4.Fem. Labor Force | −0.881 ** | −0.887** | −0.801 *** | −0.929 | 0.371 | −1.228 *** | −0.592 * |
| | (0.396) | (0.386) | (0.259) | (0.791) | (0.532) | (0.266) | (0.321) |
| L4.Grads Size | −1.796 | −2.818 | 1.470 | −7.311 *** | 2.692 | 1.974 | 1.317 |
| | (2.351) | (2.496) | (2.389) | (2.106) | (2.371) | (2.518) | (2.173) |
| L4.Diversification | 0.034 | 0.042 | 0.026 | 0.067 *** | −0.009 | −0.009 | 0.019 |
| | (0.032) | (0.032) | (0.036) | (0.020) | (0.032) | (0.033) | (0.034) |
| L4.% Grad. Fem. | 0.117 *** | 0.124 ** | 0.047 | 0.037 | −0.001 | 0.054 * | 0.043 |
| | (0.038) | (0.045) | (0.031) | (0.178) | (0.029) | (0.028) | (0.031) |
| L4.Performance gap | −0.072 | −0.037 | −0.071 ** | −0.040 | −0.029 | −0.003 | −0.058 * |
| | (0.042) | (0.040) | (0.030) | (0.064) | (0.036) | (0.046) | (0.031) |
| L4.Fertility | −7.147 *** | −8.241 *** | −7.017 *** | −11.102 * | −9.806 *** | −6.013 *** | −9.575 *** |
| | (2.501) | (2.608) | (2.251) | (5.593) | (3.129) | (1.592) | (2.570) |
| L4.Divorce rate | 1.020 ** | 0.934 * | 0.268 | 0.036 | 0.332 | 0.264 | 0.667 ** |
| | (0.460) | (0.451) | (0.325) | (1.136) | (0.270) | (0.235) | (0.255) |
| L4.Gender Equality | | −18.475 | | | | | |
| L4.Religiosity | | (0.260) | −0.231 *** | | −0.033 | −0.081 | −0.181 ** |
| | | | (0.062) | | (0.065) | (0.050) | (0.068) |
| L4.% Catholic | | | | 1.353 | | | |
| | | | | (18.047) | | | |
| L4.% Protest. | | | | 18.116 | | | |
| | | | | (14.690) | | | |
| L4% Muslim | | | | −25.867 | | | |
| | | | | (237.958) | | | |
| L4.% Jew | | | | 34.634 | | | |
| Anxiety gap | | | | (316.463) | 0.637 *** | | |
| Self−concept gap | | | | | (0.193) | 0.367 *** | |
| Self−efficacy gap | | | | | | (0.109) | 0.550 * |
| | | | | | | | (0.280) |
| No. of Obs. | 218 | 196 | 136 | 75 | 128 | 128 | 128 |
| No. of Groups | 26 | 23 | 18 | 12 | 17 | 17 | 17 |
| log−likelihood | −391.491 | −347.718 | −214.043 | −104.929 | −194.005 | −195.702 | −200.472 |
| Within R-squared | 0.337 | 0.363 | 0.408 | 0.579 | 0.470 | 0.456 | 0.414 |
| *p* value | 0.000 | 0.000 | 0.000 | 0.000 | 0.000 | 0.000 | 0.000 |

Country-level clustered standard errors in parentheses, * $p < 0.1$, ** $p < 0.05$, *** $p < 0.01$. Within-group estimates including time fixed-effects, constant terms are not reported. The fourth period lagged explanatory variables except for variables of math beliefs.

.

**Table 2.** Country-level Gender Segregation (2-digit Level Classification). Dependent variable: Dissimilarity index (2 digit-level).

|  | Cultural Values | | Math Beliefs | | |
|---|---|---|---|---|---|
|  | (1) | (2) | (3) | (4) | (5) |
| L4.Performance gap | −0.042 | −0.031 | −0.004 | 0.016 | −0.032 |
|  | (0.070) | (0.033) | (0.036) | (0.044) | (0.031) |
| L4.Fertility | −10.627 ** | −8.639 *** | −10.886 *** | −8.046 *** | −9.110 *** |
|  | (4.764) | (2.650) | (3.298) | (2.324) | (2.858) |
| L4.Gender Equality | −25.743 |  |  |  |  |
| L4.Religiosity | (22.314) | −0.195 ** | −0.024 | −0.060 | −0.155 * |
| Anxiety gap |  | (0.080) | (0.084)<br>0.477 * | (0.083) | (0.081) |
| Self−concept gap |  |  | (0.233) | 0.274 ** | |
|  |  |  |  | (0.108) | |
| Self−efficacy gap |  |  |  |  | −0.045 |
|  |  |  |  |  | (0.331) |
| No. of Obs. | 195 | 136 | 128 | 128 | 128 |
| No. of Groups | 23 | 18 | 17 | 17 | 17 |
| *p*-value | 0.000 | 0.000 | 0.000 | 0.000 | 0.000 |
| log-likelihood | −479.668 | −204.972 | −188.434 | −189.500 | −193.537 |
| Within R-squared | 0.132 | 0.480 | 0.513 | 0.505 | 0.472 |

Country-level clustered standard errors in parentheses, * $p < 0.1$, ** $p < 0.05$, *** $p < 0.01$. Within-group estimates including time fixed-effects, constant terms not reported. Fourth period lagged explanatory variables except for variables of math beliefs. The models include the full set of controls but are not reported.

## 5.2. Field and Subfield-Level Analyses

Thus far the estimates provide evidence that religiosity may partly matter to country-level horizontal gender segregation, and that gender gaps in math beliefs among young could be a more decisive factor of segregation in later education choices. This subsection seeks to identify whether religiosity and math beliefs matter to the level of gender segregation in specific fields or subfields. The models specified in Equation (4) employ the association index of either field or subfield $i$, in country $c$ in year $t$ as the LHS variable.

$$Ai_{ct} = \beta_0 + \beta_1 Religiosity_{c,t-4} + \beta_2 FieldWeight_{c,t-4} + X^J_{c,t-4}\beta_3 + \gamma_t + \alpha_c + u_{ct}$$
$$i = field(subfield); c = country; t = year$$
(5)

where $Ai_{ct}$ is the gender association of field or subfield $i$ in country $c$ and year $t$, with $\alpha_c$ and $\gamma_t$ being country and time fixed-effects. $X_{c,t-4}$ is the same set of controls as described above. To alleviate potential omitted variable bias issues, I include the proportion of graduates in each field or subfield of study in the whole of higher education in the set of control variables ($FieldWeight_{ct}$). By doing so, I also attempt to account for preferences towards specific fields of study of the whole graduate body, which might differ across countries (Alesina et al. 2013).

I first compute 9 models corresponding the 9 fields (broad classification). This step helps to narrow down the focus to estimate the impact of religiosity in specific subfields[13].

Before I review the results, it is worth noting that the $Ai_{ct}$ is a continuous variable: positive values mean over-representation of women in the field, negative values mean over-representation of men and values close to zero mean gender neutrality. Thus, to accurately interpret a significant coefficient of the regressors, one needs to know ex-ante whether the field or subfield at stake is male-labeled or female-labeled. Positive coefficients associated with the regressors in female-dominated fields would imply a positive relation with gender segregation in that it means a perpetuation of females in

---

[13] For the sake of space, all the models of the 23 subfields are not included here, but they are available upon request.

female-dominated fields. Negative values for the same coefficients would imply a negative effect on gender segregation. In considering male-dominated fields, positive (negative) values associated with the regressors would imply a negative (positive) correlation with segregation. To ease the interpretation, tables of results (Tables 3 and 4) provide the average gender-label of each field or subfield, with "F" female-domination and "M" male-domination.

Table 3 shows that religiosity seems to be associated with lower gender segregation in specifically four fields of study, namely education, science, agriculture and health and welfare. These findings might shed some light on the correlation between religiosity and lower horizontal gender segregation at national levels. All the models in Table 3 introduce the full set of controls of Equation (5), but I report the coefficients of religiosity, fertility and gender gaps in math beliefs, as they are the main contribution of the paper.

Models in Panel A (Table 3) exclude math beliefs. Fertility is not associated with a significant role in gender-labeling in any field. Religiosity is significantly associated with gender segregation in four out of the eight fields: Education and health and welfare (Columns 1 and 7), *Religiosity* enters with a negative coefficient, thus *Religiosity* is associated with reducing segregation; for science and agriculture (Columns 4 and 6), the sign is positive and the fields are male-dominated, thus *Religiosity* is associated with lower segregation in these fields.

Panels B–D (Table 3) introduce gender gaps in math anxiety, self-concept and self-efficacy, respectively. Field-level estimates tend to corroborate the finding that higher gender gaps in math beliefs are associated with higher horizontal gender segregation. Increasing gender gaps in math beliefs are persistently associated with higher male-labeling in the field of science (Column 4), but their explanatory power varies across math beliefs. However, Column (5) in Panel B associates higher math anxiety gender gaps with lower male-labeled engineering. Once gender gaps are controlled for, religiosity is still significantly associated with lower male-labeling in agriculture (Column 6) and female-labeling in health and welfare (Column 7). That is, the negative association between religiosity and gender segregation is also found in the field-level estimate. The final step in this paper is to regress Equation (5) against the $A_{ict}$ at subfield level. The results in Table 3 suggest that religiosity and gender gaps might be important for the gender-labeling of *agriculture, health and welfare*, and to a lesser extent education and science. Thus, Table 4 focus on the subfields that make up these specific fields: Science is divided into *life science*, *physical science*, *mathematics and statistics* and *computing*. Agriculture is divided into *agriculture, forestry and fishery* and *veterinary* studies. Health and welfare is divided into *health* and *social services*. Recall that education stands alone on the basis of ISCED97; it is dropped from the subfield-level analysis to avoid repetition.

Panel A in Table 4 identifies a significant link between religiosity and lower levels of male-labeling in *mathematics and statistics* (Column 3) and *agriculture, forestry and fishery* (Column 5), whereas religiosity is associated with lower female-labeling in *social services* (Column 8). These estimates suggest the same direction of the link between religiosity and segregation as previously found. When accounting for math beliefs gender gaps (Panels B–D), only the link between religiosity and *social services* remains significant at the 0.01 level. The estimates in Table 4 (Panel B) provide little evidence of a link between anxiety gaps and segregation by subfields. However, Panel C significantly associates gender disparities in math self-concept with greater segregation in *computing* and *veterinary* studies. Panel D associates math self-efficacy gaps with lower segregation in *agriculture, forestry and fishery* and *veterinary* studies.

**Table 3.** Field-level Gender Segregation. Dependent variable: Association Index (fields).

| | (1) | (2) | (3) | (4) | (5) | (6) | (7) | (8) |
|---|---|---|---|---|---|---|---|---|
| | Educ | Hum & Arts | Soc. Sci | Science | Eng. | Agri. | Health | Serv |
| **Gender-Label** | F | F | F | M | M | M | F | M |
| | | | | PANEL A: | | | | |
| L4.Fertility | −0.125 | −0.166 | 0.024 | −0.183 | 0.226 | 0.453 | −0.085 | −0.410 |
| | (0.222) | (0.186) | (0.138) | (0.187) | (0.247) | (0.350) | (0.119) | (0.359) |
| L4.Religiosity | −0.016 *** | −0.001 | 0.002 | 0.012 * | 0.005 | 0.016 ** | −0.015 ** | 0.005 |
| | (0.005) | (0.005) | (0.005) | (0.006) | (0.007) | (0.007) | (0.006) | (0.004) |
| P-value | 0.000 | 0.000 | 0.000 | 0.000 | 0.000 | 0.000 | 0.000 | 0.000 |
| No. of Obs. | 136 | 136 | 136 | 136 | 136 | 136 | 136 | 136 |
| No. of Groups | 18 | 18 | 18 | 18 | 18 | 18 | 18 | 18 |
| log-likelihood | 131.314 | 185.155 | 201.917 | 123.755 | 168.061 | 91.927 | 159.399 | 92.940 |
| Within R-squared | 0.305 | 0.304 | 0.228 | 0.276 | 0.473 | 0.272 | 0.240 | 0.267 |
| | | | | PANEL B: Math Anxiety Gender Gaps | | | | |
| Anxiety gap | 0.031 * | −0.013 | −0.002 | −0.044 ** | 0.040 *** | 0.008 | 0.039 *** | 0.057 ** |
| | (0.015) | (0.016) | (0.011) | (0.018) | (0.014) | (0.034) | (0.012) | (0.023) |
| L4.Fertility | −0.230 | −0.118 | 0.071 | −0.121 | 0.089 | 0.325 | −0.236 | −0.537 |
| | (0.218) | (0.145) | (0.149) | (0.211) | (0.202) | (0.385) | (0.154) | (0.347) |
| L4.Religiosity | −0.007 | −0.005 | −0.001 | 0.001 | 0.021 *** | 0.026 ** | −0.013 *** | 0.015 |
| | (0.009) | (0.008) | (0.008) | (0.008) | (0.006) | (0.011) | (0.004) | (0.006) |
| *p*-value | 0.000 | 0.000 | 0.000 | 0.000 | 0.000 | 0.000 | 0.000 | 0.000 |
| log-likelihood | 124.510 | 175.381 | 188.541 | 136.085 | 174.281 | 85.535 | 155.713 | 99.945 |
| Within R-squared | 0.317 | 0.338 | 0.198 | 0.425 | 0.498 | 0.271 | 0.327 | 0.342 |
| | | | | PANEL C: Math Self-concept Gender Gaps | | | | |
| Self-concept gap | 0.015 * | −0.004 | 0.006 | −0.015 * | −0.009 | −0.005 | 0.005 | 0.051 *** |
| | (0.008) | (0.007) | (0.005) | (0.008) | (0.010) | (0.021) | (0.011) | (0.017) |
| L4.Fertility | −0.067 | −0.179 | 0.088 | −0.326 * | 0.211 | 0.344 | −0.069 | −0.119 |
| | (0.202) | (0.185) | (0.164) | (0.185) | (0.195) | (0.365) | (0.132) | (0.291) |
| L4.Religiosity | −0.009 | −0.003 | 0.002 | 0.007 | 0.007 | 0.023 *** | −0.019 *** | 0.016 *** |
| | (0.007) | (0.007) | (0.007) | (0.006) | (0.009) | (0.006) | (0.006) | (0.005) |
| *p*-value | 0.000 | 0.000 | 0.000 | 0.000 | 0.000 | 0.000 | 0.000 | 0.000 |
| log-likelihood | 123.322 | 174.441 | 189.128 | 131.181 | 165.998 | 85.529 | 149.416 | 106.066 |
| Within R-squared | 0.304 | 0.328 | 0.205 | 0.379 | 0.429 | 0.270 | 0.258 | 0.402 |
| | | | | PANEL D: Math Self-efficacy Gender Gaps | | | | |
| Self-efficacy gap | 0.044 | 0.000 | 0.026 | −0.069 *** | 0.008 | −0.020 | 0.009 | −0.005 |
| | (0.028) | (0.028) | (0.025) | (0.021) | (0.028) | (0.042) | (0.024) | (0.050) |
| L4.Fertility | −0.266 | −0.160 | −0.025 | −0.070 | 0.213 | 0.434 | −0.122 | −0.324 |
| | (0.214) | (0.136) | (0.154) | (0.191) | (0.193) | (0.414) | (0.127) | (0.414) |
| L4.Religiosity | −0.013 * | −0.001 | 0.001 | 0.009 * | 0.010 | 0.023 *** | −0.020 *** | −0.000 |
| | (0.007) | (0.007) | (0.006) | (0.005) | (0.007) | (0.007) | (0.006) | (0.007) |
| *p*-value | 0.000 | 0.000 | 0.000 | 0.000 | 0.000 | 0.000 | 0.000 | 0.000 |
| log-likelihood | 123.267 | 174.188 | 190.117 | 133.821 | 165.054 | 85.637 | 149.198 | 93.625 |
| Within R-squared | 0.303 | 0.325 | 0.217 | 0.404 | 0.420 | 0.272 | 0.255 | 0.274 |
| *N* | 128 | 128 | 128 | 128 | 128 | 128 | 128 | 128 |
| No. of Groups | 17 | 17 | 17 | 17 | 17 | 17 | 17 | 17 |

Country-level clustered standard errors in parentheses, * $p < 0.1$, ** $p < 0.05$, *** $p < 0.01$. Within-group estimates including time fixed-effects and constant terms are not reported. Fourth period lagged explanatory variables except for variables of math beliefs. The models include the full set of controls but are not reported. Panels B-D include math beliefs and the number of clusters and observations are the same across fields. Educ (Education); Hum & Arts (Humanities and Arts); Soc. Sci (Social Sciences, Business and Law); Science (Science, Mathematics and Computing); Eng. (Engineering, Manufacturing and Construction); Agri. (Agriculture and Veterinary); Health (Health and Welfare); Serv. (Services). To ease the interpretation of the coefficients, behind the name of each field is the sample average gender label of M (male) and F (female), meaning whether the field is male-dominated or female-dominated, respectively. Recall that the dependent variable is a continuous variable ranging negative values for male-dominated fields and positive values for female-dominated fields.

**Table 4.** Subfield-level Segregation (selected subfields). Dependent variable: Association Index (subfields).

| Gender Label | Science | | | | Agriculture | | Health & Welfare | |
|---|---|---|---|---|---|---|---|---|
| | (1) | (2) | (3) | (4) | (5) | (6) | (7) | (8) |
| | Life S. | Phy S. | Math. | Comp. | Agri. | Vet. | Health | Soc. Serv. |
| | F | M | M | M | M | F | F | F |
| | | | | PANEL A | | | | |
| L4.Fertility | −0.005 | 0.009 | 0.144 | −0.338 | 0.389 | 0.322 | 0.234 ** | −0.425 |
| | (0.200) | (0.171) | (0.224) | (0.346) | (0.493) | (0.452) | (0.107) | (0.330) |
| L4.Religiosity | 0.009 | 0.001 | 0.024 ** | 0.008 | 0.018 ** | 0.015 | −0.008 | −0.034 *** |
| | (0.007) | (0.006) | (0.011) | (0.009) | (0.007) | (0.020) | (0.006) | (0.010) |
| No. of Obs. | 136 | 136 | 136 | 136 | 136 | 136 | 136 | 136 |
| No. of Groups | 18 | 18 | 18 | 18 | 18 | 18 | 18 | 18 |
| *p*-value | 0.000 | 0.000 | 0.000 | 0.000 | 0.000 | 0.000 | 0.000 | 0.000 |
| log-likelihood | 109.827 | 143.737 | 87.493 | 88.301 | 104.287 | 2.154 | 163.987 | 78.003 |
| Within R-squared | 0.309 | 0.210 | 0.239 | 0.731 | 0.262 | 0.370 | 0.332 | 0.476 |
| | | | PANEL B: Math Anxiety Gender Gaps | | | | | |
| Anxiety gap | −0.042 | 0.001 | −0.012 | −0.023 | 0.009 | −0.030 | 0.027 * | 0.029 |
| | (0.025) | (0.010) | (0.032) | (0.024) | (0.019) | (0.051) | (0.013) | (0.022) |
| L4.Fertility | 0.168 | −0.072 | 0.151 | −0.123 | 0.180 | 0.477 | 0.123 | −0.727 * |
| | (0.286) | (0.190) | (0.210) | (0.316) | (0.454) | (0.503) | (0.124) | (0.416) |
| L4.Religiosity | −0.006 | 0.004 | 0.022 | −0.001 | 0.018 * | 0.027 | −0.008 | −0.043 *** |
| *P*-value | 0.000 | 0.000 | 0.000 | 0.000 | 0.000 | 0.000 | 0.000 | 0.000 |
| | (0.007) | (0.006) | (0.015) | (0.014) | (0.010) | (0.016) | (0.004) | (0.011) |
| log-likelihood | 104.780 | 137.053 | 81.975 | 89.082 | 102.642 | 12.700 | 159.604 | 77.704 |
| Within R-squared | 0.340 | 0.240 | 0.185 | 0.751 | 0.249 | 0.453 | 0.379 | 0.528 |
| | | | PANEL C: Math Self-concept Gender Gaps | | | | | |
| Self-concept gap | 0.017 | 0.002 | −0.019 | −0.031 *** | −0.020 | 0.044 *** | −0.006 | 0.016 |
| | (0.014) | (0.010) | (0.014) | (0.010) | (0.012) | (0.012) | (0.012) | (0.010) |
| L4.Fertility | 0.109 | −0.063 | 0.030 | −0.324 | 0.176 | 0.568 | 0.170 | −0.556 |
| | (0.257) | (0.180) | (0.147) | (0.309) | (0.440) | (0.334) | (0.101) | (0.356) |
| L4.Religiosity | 0.011 | 0.004 | 0.018 | −0.005 | 0.011 | 0.049 ** | −0.016 ** | −0.045 *** |
| | (0.008) | (0.004) | (0.015) | (0.010) | (0.006) | (0.017) | (0.007) | (0.009) |
| *p*-value | 0.000 | 0.000 | 0.000 | 0.000 | 0.000 | 0.000 | 0.000 | 0.000 |
| log-likelihood | 102.345 | 137.080 | 83.245 | 92.443 | 104.165 | 14.792 | 156.630 | 77.316 |
| Within R-squared | 0.314 | 0.241 | 0.201 | 0.764 | 0.266 | 0.471 | 0.350 | 0.525 |
| | | | PANEL D: Math Self-efficacy Gender Gaps | | | | | |
| Self-efficacy gap | 0.021 | 0.013 | 0.070 | −0.040 | 0.061 ** | −0.124 ** | 0.024 | −0.063 |
| | (0.036) | (0.023) | (0.043) | (0.030) | (0.025) | (0.047) | (0.017) | (0.044) |
| L4.Fertility | −0.040 | −0.111 | −0.139 | −0.076 | −0.016 | 0.812 | 0.103 | −0.438 |
| | (0.226) | (0.200) | (0.160) | (0.348) | (0.436) | (0.532) | (0.104) | (0.330) |
| L4.Religiosity | 0.006 | 0.004 | 0.028 ** | 0.003 | 0.018 ** | 0.030 * | −0.012 * | −0.054 *** |
| | (0.007) | (0.005) | (0.012) | (0.011) | (0.006) | (0.016) | (0.006) | (0.008) |
| *p*-value | 0.000 | 0.000 | 0.000 | 0.000 | 0.000 | 0.000 | 0.000 | 0.000 |
| log-likelihood | 101.141 | 137.232 | 84.019 | 88.948 | 104.892 | 14.668 | 157.020 | 78.043 |
| Within R-squared | 0.301 | 0.243 | 0.211 | 0.751 | 0.275 | 0.470 | 0.354 | 0.531 |
| No. of Obs. | 128 | 128 | 128 | 128 | 128 | 128 | 128 | 128 |
| No. of Groups | 17 | 17 | 17 | 17 | 17 | 17 | 17 | 17 |

Country-level clustered standard errors in parentheses, * $p < 0.1$, ** $p < 0.05$, *** $p < 0.01$. Within-group estimates including time fixed-effects and constant terms are not reported. Fourth period lagged explanatory variables except for variables of math beliefs. The models include the full set of controls but are not reported. Panels B-D include math beliefs and the number of clusters and observations are the same across subfields. Life S. (Life Science); Phys. S. (Physical Science); Maths. (Mathematics and statistics); Comp. (Computing); Agri. (Agriculture, forestry and fishery); Vet. (Veterinary); Soc. Serv. (Social Services). To ease the interpretation of the coefficients, behind the name of each subfield is the sample average gender label of M (male) and F (female), meaning whether the subfield is male-dominated or female-dominated, respectively. Recall that the dependent variable is a continuous variable ranging negative values for male-dominated fields and positive values for female-dominated fields.

## 6. Conclusions

Persisting levels of gender segregation across fields of study in Western countries seem at odds with the increase in female participation in higher education. This observation is particularly puzzling against the backdrop of affirmative action, anti-discrimination policies and gender-egalitarian ideals in developed countries. The literature highlights individual factors (gender gaps in preferences and foreseeing family obligations) and external factors (economic structure, institutions, discrimination) as causes of gender segregation. This paper studies whether cultural values, in particular gender equality and religion, play a role in horizontal gender segregation in higher education.

I construct a panel dataset with information on gender segregation indices at national level, at 9-field level and at 23-subfield level for 26 OECD countries for 1998–2012. I link this data with two focal cultural traits: Gender equality, measured on the basis of the Gender Equality measure (IDEA), and religiosity, taken from the World Value Survey. I propose fixed-effects models that control for potential segregative factors such as economic structural change, labor market and educational systems features. The estimates fail to associate gender (in)equality measures with a significant role in horizontal gender segregation. By contrast, religiosity is significantly associated with lower levels of horizontal gender segregation.

I expand the models seeking to control for gender gaps in math beliefs developed during the youthhood. Using two waves of data taken from PISA surveys, I find a contemporaneous association between gender gaps in anxiety, self-concept and self-efficacy with higher gender segregation of graduates across fields of study. These gaps seem to be stronger predictors of national-level gender segregation than religiosity. Field and subfield-levels analyses pinpoint to a robust association between religiosity and lower segregation levels in the fields of *agriculture* and *health and welfare*, and more specifically in the subfield of *social services*.

From a policy viewpoint, the role of religiosity may be controversial. However, the findings regarding gender gaps in math beliefs tend to indicate that efforts to close gaps between boys and girls might enhance a more gender-equal distribution across fields of study in higher education. Nevertheless, it should be stressed that the findings above are based on macro-level data on segregation, and should be taken with caution. Two natural ways to extend this paper would be first to scrutinize whether there is any link between cultural traits and vertical segregation, i.e., gender segregation at different attainment levels within higher education; and second to expand the gender gaps in ability perceptions among young people into other dimensions, such as reading and science.

**Funding:** This research was funded by Institute for New Economic Thinking - Young Scholar Initiative, MINECO Spanish Ministry of Economy and Competitiveness and Gender Studies Ph.D. Theses Grant, University of the Basque Country.

**Conflicts of Interest:** The authors declare no conflict of interest.

## Appendix A. Tables in Appendices

**Table A1.** Fields and Subfields Classification (ISCED 1997) 1 digit-level. 2 digit-level.

| Education | Teacher Training and Education Science |
|---|---|
| | Arts |
| Humanities and arts | Humanities |
| Social Sciences, business and law | Social and behavioral science |
| | Journalism and Information Business and Administration Law |
| Science | Life science |
| | Physical science Mathematics and statistics Computing |
| | Engineering and engineering trades |
| Engineering, manufacturing and construction | Manufacturing and processing Architecture and building |
| Agriculture | Agriculture, forestry and fishery Veterinary |
| | Health |
| Health and welfare | Social services |
| | Personal services |
| Services | Transport services Environmental protection Security services |
| Not known or unspecified | Not known or unspecified |

**Table A2.** Summary Statistics of Explanatory Variables.

| | Mean | Std. Dev. | Min. | Max. | N |
|---|---|---|---|---|---|
| Gender Equality (IDEA) | 0.789 | 0.123 | 0.31 | 1 | 196 |
| Religiosity | 22.138 | 15.799 | 7.532 | 75.78 | 168 |
| % Jew | 0.746 | 1.534 | 0.052 | 7.378 | 168 |
| % Catholic | 36.032 | 29.524 | 0.157 | 94.400 | 168 |
| % Protestant | 22.39 | 23.451 | 0.157 | 84.117 | 168 |
| % Muslim | 7.58 | 23.584 | 0.066 | 98.886 | 168 |
| Pop. Density | 142.32 | 132.518 | 2.734 | 505.562 | 218 |
| Fem. Labor Force | 44.879 | 2.65 | 29.186 | 48.452 | 218 |
| % Services | 67.321 | 7.36 | 49.171 | 82.964 | 218 |
| % Prof. Female | 49.424 | 7.415 | 30.51 | 64.707 | 218 |
| Size Grads | 11.569 | 1.471 | 5.823 | 15.012 | 218 |
| Diversification | 19.1 | 16.042 | 0.04 | 60.004 | 218 |
| % Graduates Fem. | 57.254 | 5.673 | 25.391 | 67.5 | 218 |
| Performance gap | 4.984 | 7.413 | −21.05 | 21.36 | 218 |
| Divorce rate | 2.167 | 0.687 | 0.4 | 3.8 | 218 |
| Fertility | 1.594 | 0.29 | 1.076 | 2.23 | 218 |
| Marri. Age (females) | 28.339 | 2.048 | 23.3 | 32.8 | 218 |
| Field weight | 0.118 | 0.097 | 0.000 | 0.463 | 970 |
| Subfield weight | 0.045 | 0.053 | 0.000 | 0.32 | 2556 |
| Anxiety gap | 5.32 | 4.726 | −5.042 | 14.174 | 50 |
| Self-concept gap | 21.51 | 9.658 | 4.493 | 41.84 | 50 |
| Self-efficacy gap | 9.14 | 2.899 | 3.159 | 15.783 | 50 |

Sample of Countries (Columns 1, Table 1): Austria, Belgium, Canada, Czech Republic, Denmark, Finland, France, Germany, Greece, Hungary, Iceland, Ireland, Italy, Republic of Korea, Luxembourg, Netherlands, Norway, Poland, Portugal, Slovak Republic, Spain, Sweden, Switzerland, Turkey, United Kingdom, United States.

Sample of Countries (Data on WVS and math beliefs): Canada, Czech Republic, Finland, France, Germany, Hungary, Italy, Korea, Rep., Netherlands, Norway, Poland, Slovak Republic, Spain, Sweden, Switzerland, Turkey, United Kingdom (not in PISA), United States.

**Table A3.** Data Sources.

| Variable | Description | Data Source |
|---|---|---|
| Population Density | Number of people per square kilometer | World Bank data |
| Female Labor Force | Female labor force participation rate | ILOSTAT database |
| % Service Economy | Share of employment in service sector to total employment using the International Standard Classification of Occupations (ISCO-88) | |
| % Prof. Female | Share of females in the occupational status of "professionals" (ISCO-88: group 2) | |
| Size Grads | Share of total graduates in higher education to total population in percentages | OECD Educa- tion Database, World Bank |
| % Graduates Fem. | Share of females in total graduates in higher education | OECD Education Database |
| Performance gap | Female to male ratio of mean scores in PISA, TIMSS and PIRLS international tests from Quality of Education Database | Altinok et al. (2014) |
| Religiosity | Share of WVS respondents who say that "*God is important in my life*" equal to 10 on a 0 to 10 scale that | World Value Survey |
| Gender Equality (GE) | Measure of gender equality in participation in civil society organizations and politics and education (Skaaning 2017) | International IDEA |
| Divorce rate | Number of divorces during the year per 1000 people OECD Family | Database |
| Fertility | Total number of births per woman World Bank | |

**Table A4.** Cross-correlation table.

| | PD | Ser | Prof | FL | Grad | Diver | GFem | PG | Fert | Div | Cath | Prot | Mus | Jew | Rel | GE |
|---|---|---|---|---|---|---|---|---|---|---|---|---|---|---|---|---|
| Ser | 0.017 | | | | | | | | | | | | | | | |
| Prof | −0.281 | −0.367 | | | | | | | | | | | | | | |
| FL | −0.161 | 0.443 | 0.317 | | | | | | | | | | | | | |
| Grad | 0.066 | −0.048 | 0.282 | 0.232 | | | | | | | | | | | | |
| Diver | 0.242 | 0.082 | −0.420 | −0.333 | 0.044 | | | | | | | | | | | |
| GFem | −0.279 | 0.064 | 0.491 | 0.471 | 0.295 | −0.519 | | | | | | | | | | |
| PG | 0.354 | 0.073 | −0.216 | −0.128 | −0.314 | 0.300 | −0.280 | | | | | | | | | |
| Fert | −0.281 | 0.409 | −0.219 | −0.145 | 0.100 | 0.069 | −0.028 | −0.242 | | | | | | | | |
| Div | 0.131 | 0.217 | −0.082 | 0.428 | −0.014 | 0.124 | 0.016 | −0.066 | −0.025 | | | | | | | |
| Cath | 0.001 | −0.423 | 0.482 | 0.081 | 0.173 | −0.230 | 0.156 | 0.178 | −0.619 | −0.417 | | | | | | |
| Prot | −0.184 | 0.415 | −0.204 | 0.398 | 0.025 | −0.096 | 0.137 | −0.130 | 0.186 | 0.276 | −0.458 | | | | | |
| Mus | −0.095 | −0.651 | −0.389 | −0.790 | −0.287 | 0.263 | −0.539 | −0.133 | 0.576 | −0.446 | −0.302 | −0.303 | | | | |
| Jew | −0.330 | 0.277 | 0.333 | 0.228 | −0.001 | −0.154 | 0.171 | 0.107 | 0.297 | 0.505 | −0.118 | 0.379 | −0.114 | | | |
| Rel | −0.243 | −0.321 | 0.169 | −0.560 | 0.107 | 0.040 | −0.252 | −0.164 | 0.389 | −0.323 | 0.099 | −0.311 | 0.684 | 0.031 | | |
| GE | −0.131 | 0.665 | 0.076 | 0.748 | 0.023 | −0.221 | 0.443 | 0.057 | 0.003 | 0.327 | 0.012 | 0.452 | −0.598 | 0.167 | −0.633 | |
| Anx | −0.108 | 0.323 | −0.388 | 0.209 | −0.266 | 0.098 | −0.090 | 0.009 | 0.122 | 0.110 | −0.177 | 0.340 | −0.376 | 0.140 | −0.375 | 0.399 |
| Con | −0.002 | 0.285 | −0.511 | 0.120 | −0.378 | 0.142 | −0.257 | 0.184 | 0.080 | 0.169 | −0.231 | 0.305 | −0.152 | 0.268 | −0.099 | 0.244 |
| Effi | 0.212 | 0.451 | −0.509 | 0.247 | −0.350 | 0.010 | −0.163 | 0.315 | 0.147 | 0.138 | −0.439 | 0.527 | −0.322 | 0.005 | −0.561 | 0.489 |

PD (Pop. density); Ser (% Services); Prof (% Prof. Fem.); FL (Fem. Labor Force); Grad (Grads Size); Diver. (Diversif.); GFem (% Grads Female); PG (Performance gap); Fert (Fertility); Div (Divorce); Cath (% Catholic); Prot (% Protest.); Mus (% Muslims); Jew (% Jew); Rel (Religiosity); GE.

## Appendix B. PISA Assessment of Math Affinities

**Table A5.** Math Anxiety PISA Questions.

| Question | Boys | Girls | Girls − Boys |
|---|---|---|---|
| I often worry that it will be difficult for me in mathematics classes | 56.37 | 62.94 | 7.45 |
| I get very tense when I have to do mathematics homework | 28.05 | 31.99 | 3.94 |
| I get very nervous doing mathematics problems | 28.47 | 32.24 | 3.77 |
| I feel helpless when doing a mathematics problem | 29.25 | 34.99 | 5.74 |
| I worry that I will get poor (grades) in mathematics | 57.79 | 64.41 | 6.61 |

**Table A6.** Math Self-Concept PISA Questions.

| Question | Boys | Girls | Boys − Girls |
|---|---|---|---|
| I am just not good at mathematics (strongly disagree or disagree) | 63.26 | 52.27 | 11.11 |
| I get good grades in mathematics | 60.20 | 54.60 | 5.64 |
| I learn mathematics quickly | 58.69 | 22.92 | 40.10 |
| I have always believed that mathematics is one of my best subjects | 43.56 | 15.86 | 29.76 |
| In my mathematics class, I understand even the most difficult work | 42.76 | 15.22 | 29.03 |

**Table A7.** Math Self-Efficacy PISA Questions.

| Question | Boys | Girls | Boys − Girls |
|---|---|---|---|
| Using a train timetable to work out how long it would take to get from one place to another | 82.99 | 77.67 | 5.31 |
| Calculating how much cheaper a TV would be after a 30% discount | 84.32 | 75.98 | 8.35 |
| Calculating how many square metres of tiles you need to cover a floor | 75.77 | 61.43 | 14.34 |
| Understanding graphs presented in newspapers | 81.15 | 76.27 | 4.88 |
| Solving an equation like 3x + 5 = 17 | 83.8 | 85.2 | −1.40 |
| Finding the actual distance between two places on a map with a 1:10 000 scale | 67.44 | 48.36 | 19.08 |
| Solving an equation like 2(x + 3) = (x + 3)(x − 3) | 70.79 | 71.65 | −0.86 |
| Calculating the petrol-consumption rate of a car | 68.25 | 44.82 | 23.43 |

**Table A8.** Cross-correlation table.

| Variables | Anxiety Gap | Self-Concept Gap |
|---|---|---|
| Self-concept gap | 0.816 | |
| Self-efficacy gap | 0.388 | 0.383 |

## Appendix C

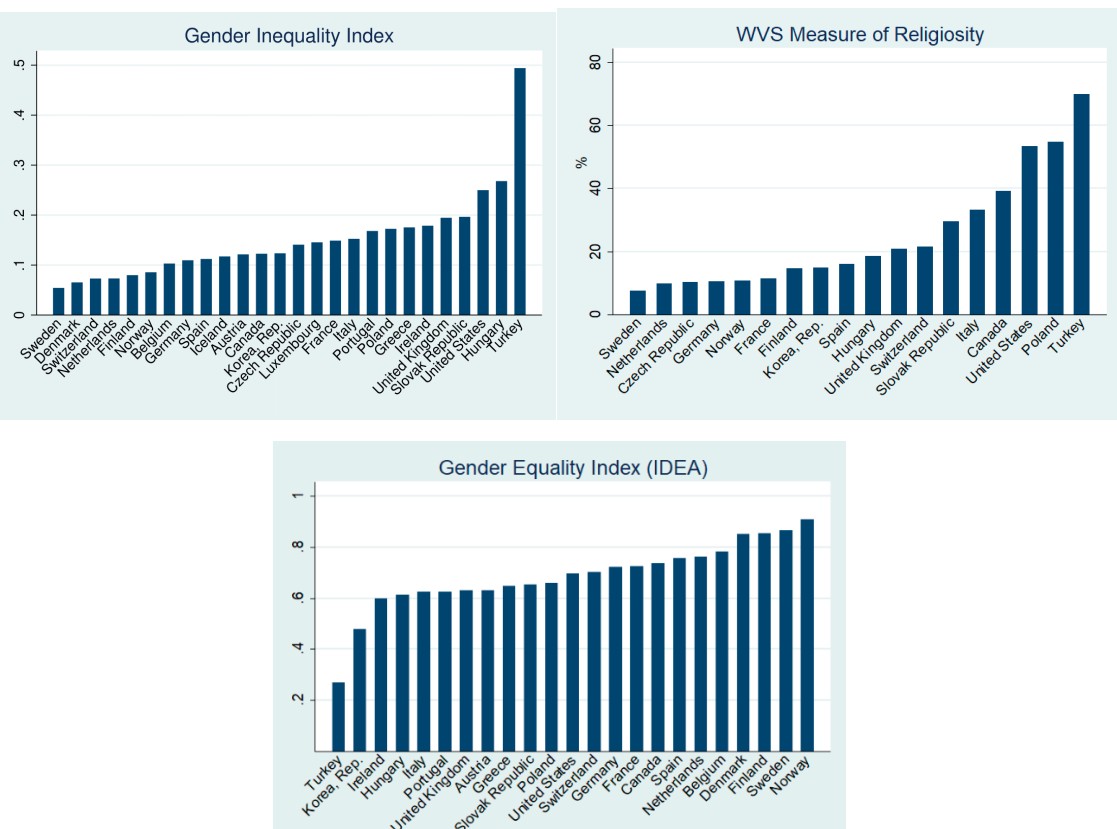

**Figure A1.** Gender-related Social Norms.

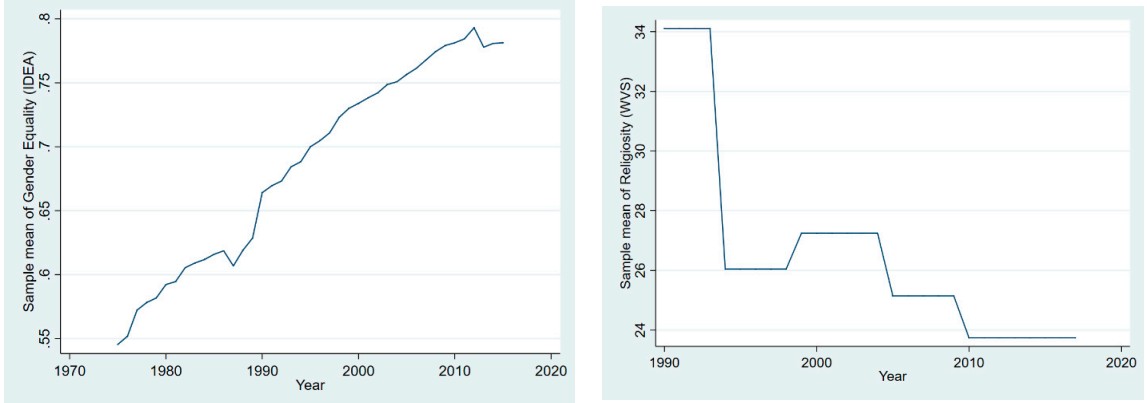

**Figure A2.** Evolution of Gender Equality and Religiosity.

## Appendix D

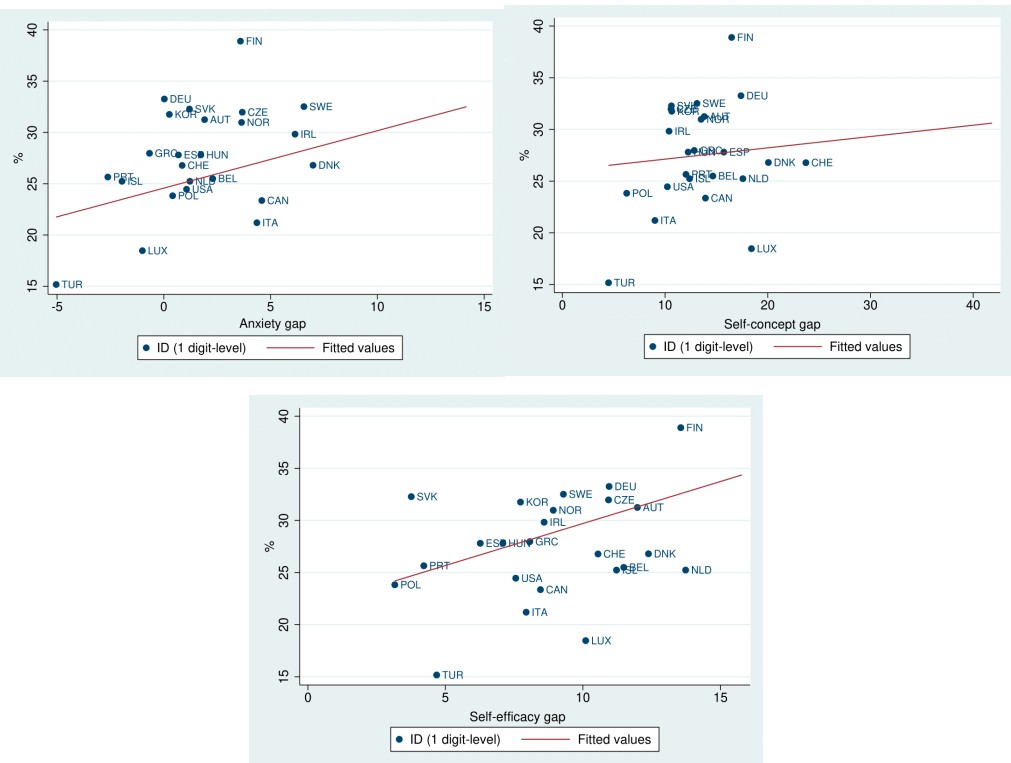

**Figure A3.** Gender Segregation and Math Self-reported Belief.

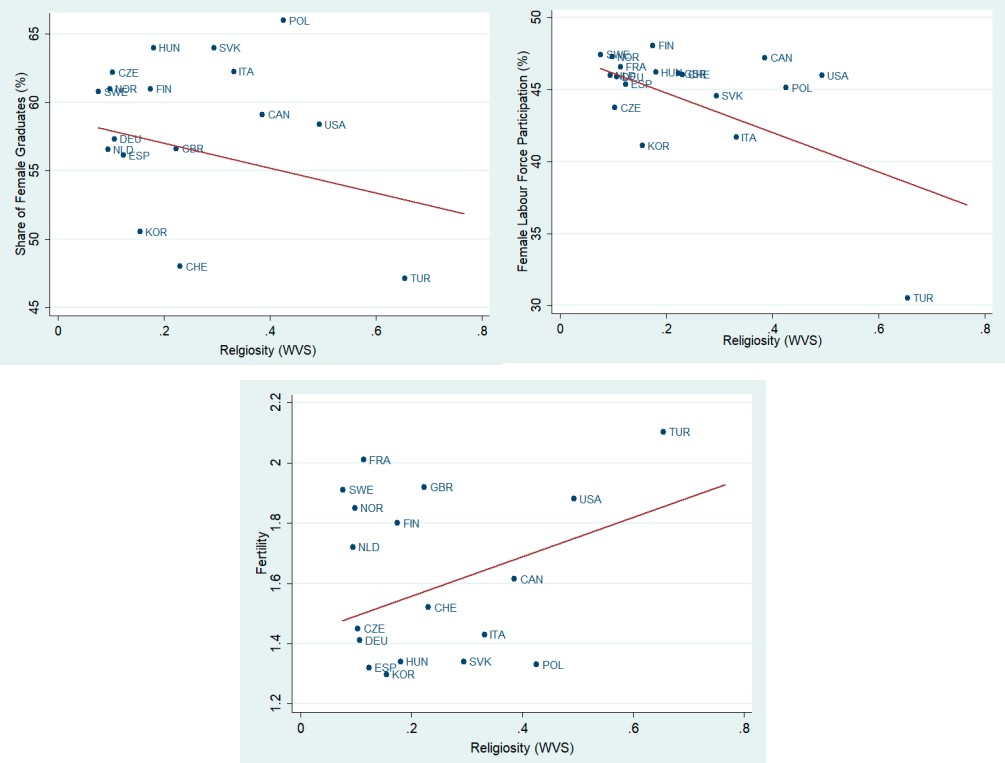

**Figure A4.** Women in the LM and HE and Religiosity.

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
