# Peer review of "Graduates’ Opium? Cultural Values, Religiosity and Gender Segregation by Field of Study"

_socsci, doi:10.3390/socsci9080135_

Round 1

Reviewer 1 Report

COMMENTS TO AUTHORS:

This paper explored the relationship between cultural values and gender distribution across fields of study in higher education and concluded that persisting levels of gender segregation across fields of study in Western countries seem at odds with the increase in female participation in higher education. I do have some comments as listed below in the order noted.

Comment 1:

The quality of the data set is very important especially data sets for 26 OECD countries for 1998-2012. For this reason, please clarify the included criteria and excluded criteria of sample collection in the Methods section and please provide a flowchart immediately at the subsection of Data Collection.

Comment2:

Please clarify and define each of the dependent variables: dissimilarity index and association index in the subsection of Study Outcomes.

Comment 3:

Please provide the P values in Tables 1-4.

Author Response

I would like to express my gratitude to this reviwer for the comments on my paper. Regarding comment #1 on the selection criteria, this was based on the availability of data and its quality. The database constructed for this paper draws on the OECD Education Database, for which some OECD countries is not available. I am not sure about what input might add to the paper a flowchart on this regard. The paper has already a substantial number of graphs with crucial information on the descriptive statistics and correlations. Nonetheless, the manuscript now includes the justification of the countries included (line 149). 

As regards comment #2, I believe that both indices are well explained already in the manuscript. Dissimilarity index and Association index are defined and their respective formulas are provided in pages 4 and 6. More importantly, the main references are provided, as well as the advantages and shortcomings of both indices are discussed in the “Data on Gender Segregation” section.

Finally, the tables now contain the p-values of the models.

Reviewer 2 Report

This paper studies whether cultural values, in particular gender equality and religion, play a role in gender segregation in  tertiary study fields. For this purpose the author constructed a novel and rich data set of macro-level data of OECD countries over a period of about 20 years. This is a very interesting question and the data set is promising in its contribution to the investigation. In the design the author has covered all the main factors that may affect segregation. In terms of the results, I found especially interesting the relationship between segregation and fertility and divorce rates.

Here are a few comments to the research design:

  1. IT would be helpful to have some more detailed description on the study field categories and how they are harmonized across countries.
  2. I wonder if the fact that many of the coefficients are not statistically significant may be a result of the seemingly significant overlap between explanatory variables. For instance, female LFP is included in the GII (or some similar labor market measure) as well as an explanatory variable on its own
  3. It would be helpful to see how much variation over time there is in the measures of religiosity and gender equality. I worry that given that they may change slowly over time, their effect might be swallowed up by the fixed effects.

Some editing comments:

  • The description of the various data sources and explanatory variables can be condensed substantially- it will make for an easier read.
  • Some references are out of date- they appear in the bibliography as working papers while they have all ready been published- e.g. Gelbgeiser and Friedman-Sokuler (a).

Author Response

I would like to thank the reviewer for her comments on my manuscript. All these comments were fundamental to improve the manuscript, and specially comment #2 on the overlapping in the model including the Gender Inequality index and the female labour force participation simultaneously. 

Regarding comment #1, Table A1 of Appendix A provides the classification of fields and subfields used in the research, namely ISCED 1997. OECD Education Database is supposed to provide harmonized data across countries. Nonetheless, this harmonization might encounter crucial limitations so long as certain degrees might not be available in some countries. This concern is now begged in the manuscript (line 156).

As said before, comment #2 raises an important issue that I now consider in the manuscript. In the previous version of the manuscript, I used the United Nations Gender Inequality Index (GII) to be able to draw parallels with other papers on the context of labour economics and alternatively used the gender equality index provided by the International Institute for Democracy and Electoral Assistance (IDEA). I did not realize about how the construction of GII imposes this problem in the estimation, so I am forced to eliminate the use of GII as a proxy of cultural values in the current version of the manuscript. Thus, the IDEA Gender Equality index is now the main proxy of cultural value related to gender norms. I am very thankful to the reviewer for pointing out this since it imposes a severe multicollinearity issue. In any case, it should be noted that the Gender Equality index by IDEA is available for a great time span and on a yearly basis thant the GII is only available for 2000, 2005, 2010, 2011 and 2012).

Regarding comment #3, the manuscript now includes Figure B2 in Appendix B that provides a graph of the evolution of the sample mean of gender equality (IDEA) and the measure of religiosity collected from the World Value Survey.

To compile with the editing suggestions, I reduced substantially the description of the control variables. Specifically, the subsection on Control Variables that regard to the labour market and education. The explanation of marriage market covariates is still in the manuscript since I feel it needs to be considered as it is one of the potential contributions of the paper.

I finally would like to thank this reviewer for checking the references. I review them accordingly and made the corresponding changes.